# Strontium and Zinc Co-Doped Mesoporous Bioactive Glass Nanoparticles for Potential Use in Bone Tissue Engineering Applications

**DOI:** 10.3390/nano14070575

**Published:** 2024-03-26

**Authors:** Parichart Naruphontjirakul, Meng Li, Aldo R. Boccaccini

**Affiliations:** 1Biological Engineering Program, Faculty of Engineering, King Mongkut’s University of Technology Thonburi, Bangkok 10140, Thailand; 2Department of Materials Science and Engineering, Institute of Biomaterials, University of Erlangen-Nuremberg, 91058 Erlangen, Germany; meng.li@fau.de (M.L.); aldo.boccaccini@fau.de (A.R.B.)

**Keywords:** antibacterial activity, bioactive glass nanoparticle, bioactivity, mesoporous, intracellular delivery

## Abstract

Mesoporous bioactive glass nanoparticles (MBGNs) have attracted significant attention as multifunctional nanocarriers for various applications in both hard and soft tissue engineering. In this study, multifunctional strontium (Sr)- and zinc (Zn)-containing MBGNs were successfully synthesized via the microemulsion-assisted sol–gel method combined with a cationic surfactant (cetyltrimethylammonium bromide, CTAB). Sr-MBGNs, Zn-MBGNs, and Sr-Zn-MBGNs exhibited spherical shapes in the nanoscale range of 100 ± 20 nm with a mesoporous structure. Sr and Zn were co-substituted in MBGNs (60SiO_2_-40CaO) to induce osteogenic potential and antibacterial properties without altering their size, morphology, negative surface charge, amorphous nature, mesoporous structure, and pore size. The synthesized MBGNs facilitated bioactivity by promoting the formation of an apatite-like layer on the surface of the particles after immersion in Simulated Body Fluid (SBF). The effect of the particles on the metabolic activity of human mesenchymal stem cells was concentration-dependent. The hMSCs exposed to Sr-MBGNs, Zn-MBGNs, and Sr-Zn-MBGNs at 200 μg/mL enhanced calcium deposition and osteogenic differentiation without osteogenic supplements. Moreover, the cellular uptake and internalization of Sr-MBGNs, Zn-MBGNs, and Sr-Zn-MBGNs in hMSCs were observed. These novel particles, which exhibited multiple functionalities, including promoting bone regeneration, delivering therapeutic ions intracellularly, and inhibiting the growth of *Staphylococcus aureus* and *Escherichia coli*, are potential nanocarriers for bone regeneration applications.

## 1. Introduction

The inherent capacity of bone for dynamic self-repair and regeneration without forming fibrous scar tissue via the endochondral pathway is widely recognized. However, this ability can be limited, leading to impaired bone regeneration and compromised regenerative processes, for example, in various scenarios such as post-tumor removal or major trauma, osteoporosis in patients, and accidents [1]. The gold standard for bone damage treatment is an autograft harvested from the patient, as it will contain osteoinductive growth factors, osteogenic cell sources, and an osteoconductive scaffold [2,3]. Primary challenges in the transplantation of natural bone grafts include the constrained supply of healthy bone from patients [4,5] and the occurrence of morbidity at donor sites [5,6]. Due to these limitations, allografts and xenografts are used instead. Neither allografts nor xenografts are harvested from the bone of the patient; therefore, both disease transmission and rejection risks can be high [4,7,8]. To overcome these restrictions, synthetic bone graft substitutes have been developed to replicate the characteristics of human bone, aiming to mitigate issues associated with natural grafts [2,7,9,10]. Natural and synthetic biomaterials are considered in using tissue engineering and regenerative medicine approaches [9].

Bioceramics, such as calcium sulfate, calcium phosphate (CaP), hydroxyapatite (HA), and bioactive glasses (BGs), constitute the majority, accounting for 60% of the synthetic bone graft substitutes accessible in the global market [11]. BGs are used clinically for dental and orthopedic applications because of their biocompatibility, nontoxicity, noninflammatory effects, angiogenesis, and osteoconductive and osteoinductive properties. BGs exhibit bioactivity by developing the formation of an apatite layer similar to bone on their surface upon exposure to physiological solutions [11,12,13,14,15]. Moreover, BGs elicit a distinct biological response, exhibiting a faster bonding with living tissue through a hydroxycarbonate apatite (HCA) layer compared to other bioceramics [16,17,18,19]. Nanoparticles (NPs) are employed clinically as nanocarriers to deliver therapeutic agents. The dimensions and shape of NPs significantly influence their internalization and localization. Therefore, BGs developed at the nanoscale can increase bioactivity due to their high specific surface area that accelerates possible interactions with biological molecules [1,20,21].

Recently, there has been notable interest in mesoporous silica nanoparticles (MSNs) owing to their diverse biomedical applications.

The properties of MSNs include biocompatibility, exhibiting minimal cytotoxicity and immunogenicity; a highly ordered porous structure with a high surface area; tunable particle sizes, pore architectures, distribution, and interconnectivity; and the ability to be tailored to optimize functional groups on their surfaces [22,23,24,25,26,27]. The tunability of pore size in MSNs is a key advantage of these materials. In order to form the pore pattern, both natural and charged surfactants, such as non-ionic triblock polymers (Pluronic F127 and Pluronic P123) and cationic surfactants (cetyltrimethylammonium bromide, CTAB), are used to control the pore and particle sizes [23].

Synthesizing MSNs involves carefully controlling the conditions of a chemical reaction to form silica structures with ordered and interconnected nanoscale pores. MSNs are usually synthesized through the sol–gel process. “Sol” denotes a stable suspension comprising solid particles dispersed in a liquid medium, while “gel” signifies a three-dimensional network structure resulting from the aggregation and cross-linking of colloidal primary particles within a liquid medium [28,29]. MSNs are useful materials with numerous applications, but they have challenges and limitations due to their composition. MSNs exhibit inherent bioactivity due to the essential role of Si in bone formation. Calcium (Ca) has been incorporated into MSNs to improve their bioactivity, leading to mesoporous bioactive glass nanoparticles (MBGNs) [27].

The specific biological properties of MBGNs can be tailored by adding therapeutically active substances, such as ions [30,31]. Moreover, the ordered pore structure of MBGNs in the range of 2–50 nm can accelerate the ability to form bone-like apatite on the particles’ surface. The microemulsion-assisted sol–gel method combined with a cationic surfactant (CTAB) is widely used to synthesize MBGNs [32]. After removing the surfactant used as the pore template, a mesopore pattern remains in the MBGNs. A microemulsion composed of the correct mixture of oil, water, and surfactant forms a stable dispersion, avoiding droplet aggregation [33]. It has been reported that the combination of ethyl acetate and CTAB could transform spherical micelles into rod-like particles under basic conditions in an oil-in-water (O/W) microemulsion-assisted sol–gel system, ultimately resulting in the creation of mesoporous structures [31].

MBGNs exhibit considerable promise in biomedical applications owing to their adjustable features, including size, morphology, pore size, pore pattern, and surface functionalization. These attributes enable the controlled release of therapeutic ions or molecules, thereby eliciting specific cellular responses [34]. To optimize the effectiveness of MBGNs in biomedical contexts, it is imperative to tailor their degradability and bioactivity by incorporating therapeutic cations into their network structure. Specifically, MBGNs are being explored for the simultaneous stimulation of osteogenesis and angiogenesis through flexible modifications to the network structure. Bioactive cations such as strontium (Sr) [35,36], zinc (Zn) [37], manganese (Mn) [38], cerium (Ce) [39], copper (Cu) [40], and cobalt (Co) [36] have been recently incorporated into MBGNs (in the SiO_2_-CaO binary glass system). Therapeutic cations in MBGNs can influence bioactivity, degradation rate, mechanical properties, and specific biological response.

Sr was introduced into the glass structure as a network modifier by replacing a portion of CaO with SrO, leading to the enhancement of bone remodeling [41]. Sr enhanced the metabolic activity of osteoblasts, promoted cell growth, and increased alkaline phosphatase (ALP) activity [42]. Sr possess the capability to boost bone formation by promoting osteoblast function and to impede bone resorption by suppressing osteoclast activity [43,44]. Zn was doped into MBGNs to stimulate osteoblast activity, enhance bone formation, and enhance antimicrobial activity. This is particularly important in applications where the prevention of bacterial growth is desired, such as in wound healing or implant coatings. The concentration of Sr and Zn can affect the osteogenic properties of MBGNs, with an optimal range leading to the most effective bone healing outcomes [45,46,47].

Although the bioactivity of Sr- and Zn-doped MBGNs has been reported, no current research has studied the multifunctional effects of co-doping with Sr and Zn in MBGNs. The novelty of this study therefore derives from its attention to co-doping with Sr and Zn in MBGNs and using them as a potential alternative nanocarrier for bone regeneration applications. Infections during bone treatment pose a significant risk of hindering the success of reconstruction. The incidence of post-surgical infections has been documented to vary between 12% and 47% [48]. *Staphylococcus aureus* is the predominant Gram-positive bacteria linked to infections related to biofilms [49]. The hypothesis was that Sr- and Zn- doped MBGNs have the ability to internalize and localize within the cell and release the therapeutic ions to enhance osteogenic properties and antibacterial activity. Sr^2+^ and Zn^2+^ ions have great potential to combine functionalities for bone regeneration including proliferation, differentiation, and inhibiting bacterial growth. Hence, the newly developed MBGNs exhibit greater potential for promoting bone formation and addressing infections compared to alternative nanoparticles. The aims of this study were thus to synthesize MBGNs using the microemulsion-assisted sol–gel technique with modification of the pore pattern using cationic surfactants (CTAB) and to investigate the impact of Sr and Zn doped into the 60SiO_2_-40CaO binary glass system on bioactivity and cellular response.

## 2. Materials and Methods

All reagents were from Sigma-Aldrich (St. Louis, MO, USA) unless stated otherwise as follows: tetraethyl orthosilicate (TEOS), ethyl acetate, ammonium hydroxide, ethyl alcohol, calcium nitrate tetrahydrate (99%), strontium nitrate (99%), phosphate-buffered saline (PBS), paraformaldehyde, Alizarin Red S, 3-aminopropyl triethoxysilane (APTES), fluorescein 5(6)-isothiocyanate (FITC), dimethyl sulfoxide (DMSO), L-ascorbic acid, β-glycerophosphate, dexamethasone (DEX), sodium chloride (NaCl), potassium chloride (KCl), nitric acid, sodium hydrogen carbonate (Na-HCO_3_), Hexadecyltrimethylammonium bromide (CTAB, CH_3_(CH_2_)15N(Br)(CH_3_)_3_, ≥98%), di-potassium hydrogen phosphate trihydrate (K_2_HPO_4_.3H_2_O), sodium phosphate, magnesium chloride hexahydrate (MgCl_2_.6H_2_O), sodium sulfate (Na_2_SO_4_), hydrochloric acid (HCl), toluene, calcium chloride (CaCl_2_), cetylpyridinium chloride, minimum essential medium eagle alpha (α-MEM, Gibco^TM^, Bangkok, Thailand), fetal bovine serum (Gibco^TM^, Bangkok, Thailand), Antibiotic-Antimycotic (Gibco^TM^, Bangkok, Thailand), Penicillin–Streptomycin (Gibco^TM^, Bangkok, Thailand), trypsin-EDTA (Gibco^TM^, Bangkok, Thailand), 3-(4,5-dimethylthiazol-2-yl)-2,5-diphenyltetrazolium bromide (MTT, Thermo Fisher Scientific, Bangkok, Thailand), Mueller-Hinton Agar (MHA, Difco™, Bangkok, Thailand), Human mesenchymal stem cells (hMSCs: ATCC^®^ PCS-500-012, Biomedia, distributor of ATCC, Bangkok, Thailand).

### 2.1. Mesoporous Bioactive Glass Nanoparticle (MBGN) Synthesis

MBGNPs were synthesized via the modified microemulsion-assisted sol–gel process, with 1 N ammonium hydroxide serving as the base catalyst to regulate hydrolysis and polycondensation reactions, thus controlling the particle size (Figure 1). The nominal ratio of MBGN composition is reported in Table 1. Briefly, 5 g of cetyltrimethylammonium bromide (CTAB), the cationic surfactant to generate the pore pattern [50,51], was dissolved in 260 mL of pre-heated deionized water at 55 °C in a 1 L DURAN^®^ original laboratory bottle at a stirring rate of 600 rpm for 3 min. Then, 80 mL of ethyl acetate was simultaneously introduced to the solution and allowed to mix for 30 min at 55 °C. Then, 56 mL of 1 N ammonium hydroxide solution was added. After stirring for 15 min, 28.8 mL of tetraethyl orthosilicate (TEOS) was gently mixed and stirred for another 30 min, and 18.3 g of calcium nitrate tetrahydrate was then added. In this study, Sr (Sr-MBGNs), Zn (Zn-MBGNs), and Sr and Zn (Sr-Zn-MBGNs) were substituted for Ca (MBGNs) to extend the biological application. After adding TEOS for 30 min, 10.9 g of calcium nitrate tetrahydrate and 19.7 g of strontium nitrate (Sr-MBGNs); 10.9 g of calcium nitrate tetrahydrate and 13.7 g of zinc nitrate (Zn-MBGNs); and 10.9 g of calcium nitrate tetrahydrate, 4.9 g of strontium nitrate, and 6.9 g of zinc nitrate (Sr-Zn-MBGNs) were added. Following a 4 h reaction period, the colloidal particles were harvested via centrifugation at 7380 rpm for 30 min. The collected particles underwent two washes with deionized water and two washes with ethanol before being dried at 80 °C and calcined at 700 °C, with a heating rate of 2 °C/minute, for 4 h.

### 2.2. Physicochemical Characterization of MBGNs

To analyze the particle size and shape, dynamic light scattering (DLS, Horiba SZ-100V2, Horiba, Kyoto, Japan) and field emission scanning electron microscopy (FE-SEM, Auriga, Carl Zeiss with an accelerating voltage of 1 kV) were used. Furthermore, the particle size of the MBGNs was confirmed using the ImageJ software (version 1.41o, Java 1.6.0_10, Wayen Rasband, US National Institutes of Health, Bethesda, MD, USA). EDS-SEM was used to confirm the elemental components of the synthesized MBGNs. To evaluate the stability of particles in solutions, Zeta potential measurements were conducted in distilled water at pH 6.4 utilizing a Zeta sizer (Horiba SZ-100V2, Horiba, Kyoto, Japan). N_2_ adsorption/desorption isotherms were acquired employing a specialized surface area analyzer (BET, BELSORP-mini II, Bel Japan Inc., Osaka, Japan). BET multipoint analysis was performed to determine the specific surface area, with P/Po data points selected from the range of 0.05–0.35. Prior to analysis, the samples underwent degassing at 200 °C for 24 h at a heating rate of 10 °C/min [51]. Fourier transform infrared spectroscopy (FTIR; Thermo Scientific Nicolet iS5, Thermo Fisher Scientific, Waltham, MA, USA) was utilized to characterize the functional groups present in the MBGNs. FTIR spectra were obtained in attenuated total reflection (ATR) mode across a wavenumber range of 4000 to 400 cm^−1^, with a scan speed of 32 scans/min and a resolution of 4 cm^−1^. X-ray diffraction (XRD; Bruker AXS Model D8 Advance, Karlsruhe, Germany) was employed to analyze the crystalline structure of the particles. XRD patterns were collected using a Bruker AXS automated powder diffractometer with Cu Kα radiation (1.5406 A°) at 40 KV/40 mA, within the 10–70° 2θ range, with a step size of 0.02° and a dwell time of 0.5 s. X-ray fluorescence (XRF; Fisher/XUV773, Fischer Instrumentation, Worcestershire, UK) with X-ray generators operating at 20 kV in a vacuum was utilized to determine the elemental compositions within the MBGNs.

### 2.3. In Vitro Bioactivity Study

In total, 75 mg of MBGNs, Sr-MBGNs, Zn-MBGNs, and Sr-Zn-MBGNs were immersed in 50 mL of a preheated Simulated Body Fluid (SBF) solution at pH 7.4 and 37 °C, while shaking at 120 rpm, for 21 days. Upon completion of the incubation period, the particles were harvested through centrifugation at 7380 rpm for 10 min and then washed. SEM (JEOL, JSM-6610 LV, JEOL Ltd., Tokyo, Japan) was used to observe the morphological changes of the immersed MBGNs, Sr-MBGNs, Zn-MBGNs, and Sr-Zn-MBGNs. EDS-SEM (OXFORD, INCAx-act, Oxford Instruments, Abingdon, UK) was utilized to analyze the elemental composition of the incubated particles. The SBF solution was prepared according to a previously published protocol [52].

### 2.4. In Vitro Cell Viability

Human mesenchymal stem cells (hMSCs: ATCC^®^ PCS-500-012™) were routinely cultured in α-MEM supplemented with 10% (*v*/*v*) fetal bovine serum (FBS) and 100 U/mL of Antibiotic-Antimycotic (Invitrogen, Gibco^®^, Grand Island, NY, USA) at 37 °C, 5% CO_2_, and in a fully humidified atmosphere. Passages 2 and 3 of the hMSCs, maintained at a concentration of 5 × 10^4^ cells/mL, were used. To assess the cytotoxic effects of the particles on hMSC viability, an MTT colorimetric assay was performed following the manufacturer’s instructions. The monolayer of hMSCs was treated with different concentrations of particles ranging from 0 to 1000 µg/mL: 0, 10, 100, 200, 250, 300, 400, 500, 750, and 1000 µg/mL for 24 h. Metabolic cell activity was assessed using the MTT assay (0.5 mg/mL), which measures the conversion of MTT into formazan. The soluble formazan was dissolved in DMSO and then the concentration was measured using a microplate reader (Infinite^®^ 200 Tecan Austria GmbH, Grödig, Austria). at 570 and 620 nm. The percentage of relative cell viability, compared to untreated control cells, was determined by calculating the mean value ± standard error of the mean. Statistical analysis was conducted using ANOVA with Tukey’s post hoc comparison test. The experiment comprised six technical replicates (*n* = 6) and was repeated in triplicate (N = 3). Cell viability less than 70% is considered a cytotoxic effect (ISO 10993-5).

### 2.5. In Vitro Mineralization

To evaluate the calcium phosphate deposits of hMSCs, alizarin red was monitored. hMSCs were seeded in 24-well plates with a cell concentration of 5 × 10^3^ cells/mL. The basal α-MEM was used as the negative control. For positive control, an osteogenic medium was prepared by supplementing the basal medium with 10 nM of dexamethasone (DEX), 10 mM of β-glycerophosphate, and 100 µg/mL of L-ascorbic acid. The monolayer of hMSCs was exposed to media containing NPs at a concentration of 200 µg/mL (cut-off-level particle concentration with no cytotoxic). The cell culture media containing NPs were regularly replaced, twice a week, over a period of 3 weeks. During the 1st, 2nd, and 3rd weeks in culture, the cells were immobilized with 4% paraformaldehyde. The calcification of hMSCs was identified by employing 2% Alizarin Red S solution in deionized water at pH 4.2. Images were captured utilizing an inverted optical microscope (LABOMED TCM400, Los Angeles, CA, USA) with the ToupView program. To conduct the quantification assay, the stained cells were rinsed with deionized water to eliminate nonspecific staining. The stain was extracted using 10% (*w*/*v*) cetylpyridinium chloride in 10 mM of sodium phosphate buffer at pH 7.0, and absorbance was subsequently measured at 562 nm using a microplate reader (Infinite^®^ 200 Tecan, Grödig, Austria).

### 2.6. Osteogenic Differentiation

Real-time quantitative polymerase chain reaction (q-PCR) was used to assess the expression of osteogenic differentiation marker genes in hMSCs. Initially, hMSCs were seeded in 6-well plates at a concentration of 5 × 10^3^ cells/mL. Subsequently, the monolayer of hMSCs was exposed to media containing NPs at a concentration of 200 µg/mL. The medium containing particles was regularly refreshed twice a week for a duration of 3 weeks. Cells treated with MBGNs were employed as control samples. At designated time intervals (1, 2, and 3 weeks), total RNA was isolated using a Monash total RNA isolation kit (Monarch^®^ Total RNA Miniprep Kit, New England Biolabs, Ipswich, MA, USA), according to the manufacturer’s instructions. The concentrations of total RNA were assessed using a nanodrop (NANODROP 2000c Spectrophotometer, Thermo Fisher Scientific). Following that, cDNA was generated utilizing the iScript™ cDNA Synthesis Kit (Bio-Rad, Bio-Rad Laboratories, Hercules, CA, USA), following the manufacturer’s protocol. Subsequently, q-PCR analysis was carried out employing the CFX96™ Real-Time PCR Detection System (Bio-Rad, Bio-Rad Laboratories, Hercules, CA, USA) and prepared with iTaq Universal SYBR Green Supermix (Bio-Rad, Bio-Rad Laboratories, Hercules, CA, USA), following the manufacturer’s instructions. The primer pair sequences utilized in this experiment are presented in Table 2. The relative gene expression was determined using the comparative 2^−ΔΔCt^ method, with normalization to the reference gene, GAPDH. All reactions were performed in triplicate.

### 2.7. Cellular Uptake

Sr-MBGNs, Zn-MBGNs, and Sr-Zn-MBGNs were functionalized and labelled with FITC modified from [53]. First, amine groups were conjugated into particles; 200 mg of particles were dispersed in 45 mL of toluene and 5 mL of APTES at a stirring rate of 400 rpm, at 70 °C, for 18 h. Then, amine-conjugated particles were collected and washed with ethanol and DI water twice to remove the excess components. After that, collected particles were resuspended in 1 mg/mL of FITC in ethanol in an incubator shaker at 200 rpm, at 37 °C, for 4 h in dark conditions. FITC-conjugated MBGNs were collected and washed with ethanol. The MTT assay was used to investigate the effect of the FITC-conjugated MBGNs on the viability of the hMSCs. hMSCs were seeded in flat-bottomed 96-well plates (Costar, Corning Inc., Corning^®^, Corning, NY, USA) with a cell concentration of 5 × 10^4^ cells/mL and incubated at 37 °C, 5% CO_2_, and in a fully humidified atmosphere overnight to facilitate cell attachment in the monolayer. The hMSCs were treated with different concentrations of particles ranging from 0 to 250 µg/mL: 0, 1, 10, 50, 100, 200, and 250 µg/mL. After 24 h of exposure, cell viability was determined using the MTT colorimetric assay

For the cellular uptake, hMSCs at a cell density of 5 × 10^4^ cells/mL were seeded on 6-well plates and incubated at 37 °C, 5% CO_2_, and in a fully humidified atmosphere overnight to facilitate cell attachment in the monolayer. The hMSCs were treated with FITC-conjugated MBGNs at concentration of 200 µg/mL. After 24 h of exposure, the cells were washed and fixed with 4% paraformaldehyde in PBS. 4′,6-diamidino-2-phenylindole (DAPI) was used to determine the nuclei. The internalization of particles was visualized using a confocal laser scanning microscope (OLYMPUS, FV1000, Olympus Corp. Tokyo, Japan).

### 2.8. In Vitro Antibacterial Activity

Antibacterial activity was determined using the disk diffusion method. *Staphylococcus aureus* ATCC 6538 and *Escherichia coli* ATCC 8739 were obtained from the Department of Microbiology, Faculty of Science, King Mongkut’s University of Technology Thonburi, Thailand. *E. coli* and *S. aureus* isolates were grown overnight on Mueller Hinton Agar (MHA) at 37 °C. Direct colony suspensions were prepared in sterile saline to achieve a turbidity equivalent to 0.6 McFarland standard. The inoculum suspensions were streaked onto the dried surface of an MHA plate. Twenty microliters of MBGNs were dropped onto 6 mm diameter Whatman^®^ antibiotic assay disks in a sterile dish. Disks were placed aseptically onto the plates immediately. Penicillin–Streptomycin at 100 U/mL (Invitrogen, Gibco^®^, Grand Island, NY, USA) was used as a positive control. The diameter of the inhibition zones or halo zones was measured after incubation at 37 °C for 16 to 18 h.

### 2.9. Statistical Analysis

The graphs shown present the results as the mean value with the standard deviation (SD) as the error bars. All quantitative experiments were carried out at least in triplicate. Statistical analyses were performed by one-way analysis of variance (ANOVA) in R studio (R Core Team, Vienna, Austria) [54] with the appropriate post hoc comparison test (Tukey’s test). A *p*-value < 0.05 was considered significant.

## 3. Results and Discussion

### 3.1. Characterization of MBGNs

Monodispersed cation-doped MBGNs were successfully synthesized through the microemulsion-assisted sol–gel process using ammonium hydroxide as the base catalyst to control hydrolysis and polycondensation reactions to control the size of the particles. The cationic surfactant, CTAB, played an important role in modulating the interior mesoporous structure, morphology, and dispersity of the MBGNs. In O/W microemulsion droplets (ethyl acetate in water), CTAB self-assembled to form micelles that were used as a templating agent to create a mesoporous structure [32]. Therapeutic cations, including Ca^2+^, Sr^2+^, and Zn^2+^, were incorporated into the silica network through the heat treatment process. Calcination at 700 °C for 4 h was used to decompose the unreacted precursors and to create the pore structure [55].

The hydrodynamic diameter based on the diffusion of the particles was measured using dynamic light scattering (DLS; Horiba SZ-100V2, Horiba, Kyoto, Japan), as shown in Table 3. The diameter of the synthesized MBGNs was 160 ± 20 nm. The suspension stability in aqueous solution was measured using the ζ potential. Generally, ζ potentials of ±30 mV cause a repulsive force between the particles and prevent agglomeration [56]. The ζ potential of the synthesized MBGNs was −40 ± 2 mV, indicating that the MBGNs were stable in aqueous environments at pH 7.0. The surface charge of the particles affected protein adsorption and cellular interaction. According to previous reports, a positive charge on the particle surface enhances the attraction and binding of negatively charged proteins to the surface. This promotes an opsonization process, which ultimately facilitates the removal of the particles circulating in the body [57]. Moreover, the surface charge of nanoparticles can influence their cellular uptake and intracellular localization. Therefore, the presence of negatively charged MBGNs due to the presence of silanol groups might hinder the elimination of the particles from the systemic circulation.

The size and morphology of the particles were investigated using scanning electron microscopy (SEM; Figure 2). The SEM images show the spherical shape, uniformity, and homogeneity of the particles from 100 ± 20 nm (ImageJ, USA, *n* = 50). The size of the particles measured using DLS was slightly larger than the measurements using SEM. This result might be because DLS analyzes fluctuations in the scattered light caused by the random movements of the particles and the surrounding solvent molecules (Brownian motion). DLS normally provides an average hydrodynamic diameter of nanoparticles in a solution, while SEM provides detailed morphology, size, shape, and surface features of individual nanoparticles [58,59]. The pore patterns on the surface of the particles were monitored. The co-doping of therapeutic bivalent cations did not affect particle size, morphology, or pore pattern structures, as shown in Figure 2. Previous research showed that cation precursors were located on the surface of the secondary particles after drying and connected to the silica network after calcination [60]. The bivalent cations (Sr^2+^ and Zn^2+^) were successfully incorporated in the MBGNs, as shown by the SEM-EDS results (Figure 2). In general, the surface of MBGNs featured a high presence of silanol groups (Si-OH), leading to an overall negative surface charge, as indicated by the ζ-potential values (Table 3). The incorporation of Sr^2+^ and Zn^2+^ did not notably alter the ζ potential, as these divalent cations were effectively substituted with Ca^2+^ in the chemical composition and primarily integrated into the silica network. The zeta potential is influenced by the chemical composition, surface characteristics, and size of MBGNs. Biomaterials exhibiting a negative zeta potential (an electronegative surface charge) contribute to cell proliferation and facilitate new bone formation [61].

The silica structure was created via a simultaneous process involving hydrolysis and polycondensation reactions. The doped ions functioned as network modifiers and intermediates that modified the silica network through the heat treatment step [62]. The nominal composition refers to the quantity of the precursor added during the synthesis process, as outlined in Table 1. During the washing phase to remove CTAB in the preparation of MBGNs, some of the partially doped ions were eliminated. As a result, not all of the expected components were incorporated into the particles. XRF analysis was employed to identify the elemental composition of the synthesized MBGNs, validating the successful substitution of Ca with Sr and Zn within the glass network (Table 4). The SiO_2_ content of the MBGN (60SiO_2_-40CaO) binary glass system was around 80%mol, indicating that not all ions from the nominal ratio were integrated into the silica network due to the CTAB removal during the washing process. The amount of Si and Ca decreased when Sr was doped into particles from 83.1 ± 0.8 to 78.3 ± 0.6%mol. The amounts of Ca, Sr, and Zn incorporated in the particles depended on the starting nominal mol%. The inclusion of Zn resulted in a notable decrease in the quantities of Ca and Sr, suggesting that Zn has high binding efficacy to the silica network in comparison to Ca and Sr. Zn, an intermediate, had a significant impact on facilitating changes in the connectivity of the network, modifying the bioactivity, and influencing the release of ions and dissolution of the glass [63].

Previous studies have indicated that replacing larger ions with smaller ions (Zn^2+^ instead of Ca^2+^) results in the densification of the glass network, leading to a reduction in molar volume and an elevation in oxygen density. Additionally, the inclusion of ZnO as a network-forming component promotes stronger glass interconnection, thereby facilitating the formation of Si−O−Zn units [64,65]. These results indicate that the composition of the original MBNGs (60SiO_2_-40CaO) can be modified with Sr and Zn using the microemulsion-assisted sol–gel and subsequent calcination process. No studies have explored the impact of co-doping with Sr and Zn in MBGNs on osteogenic properties and antibacterial activity yet. Previous studies reported that Zn-containing MBGNs (8ZnO, %mol and 23ZnO, %wt) stimulated osteogenic, anti-inflammatory, and anti-bacterial activities [37,66,67]. Sr-containing MBGNs (0.5–10 SrO, %mol) increased biodegradability, biocompatibility, and promoted osteoblast bone-forming activity [68,69]. Thus, these particles with optimal Sr and Zn dopants can enhance bioactivity and osteoblast activity.

FTIR spectroscopy was utilized to analyze the structural changes within synthesized MBGNs. The FTIR spectra indicated that the chemical bonding of the silica networks in the MBGNs was identical. The characteristic bioactive glass transmittance peaks were observed, as shown in Figure 3a. The presence of a peak within the range of 450–460 cm^−1^ corresponds to the Si-O-Si bending vibration of amorphous structures. Peaks at 800 and 1000–1200 cm^−1^ were attributed to symmetric and asymmetric Si-O-Si stretching modes, respectively, signifying the presence of SiO_4_ tetrahedrons, the basic building block of the glass network [70]. When Zn was doped into the MBGNs, a weak peak emerged at 930–950 cm^−1^, attributed to the Si-O non-bridging oxygen (NBO) stretching mode. In the MBGNs doped with Zn and Sr, the peak shifts from 930–950 to 950–970 cm^−1^. These findings suggest the integration of Zn and Sr; the peak shift indicates a modification in the silica network formation due to the dissociation of the Si-O-Si bond. The XRD spectra of MBGNs show a broad halo band located at 2θ = 20° − 30° that relates to the amorphous structure characteristic of the glassy phase, indicating that Sr and Zn were effectively incorporated into the MBGNs while the amorphous structure was maintained (Figure 3b). The combination of Sr and Zn (Sr-Zn-MBGNs) broadened the halo peak, implying a disordered arrangement at the nanoscale level. The structure of the glass was impacted by the substitution of different modifier cations or intermediate ions, which occurred due to differences in their sizes and properties. When modifier cations with identical valences but varying sizes were substituted (such as Sr^2+^ in place of Ca^2+^), the silicate network expanded or became less tightly packed, increasing solubility and bioactivity. Additionally, the presence of the intermediate ion (Zn^2+^) reduced the likelihood of crystallization while maintaining the connectivity of the network [71].

Nitrogen adsorption measurements were used to investigate the surface area, pore size, and pore volume of the MBGNs. The presence of a Type IV isotherm with a hysteresis loop observed between 0.4 and 0.9 P/P0 indicated the presence of mesopores, typically ranging from 2 to 50 nm (IUPAC classification) [72]. The synthesized MBGNs represented the Type IV isotherm (Figure 4), indicating a mesopore pattern. The textural characteristics of MBGNs are presented in Table 5. The pore size diameter distributions were comparable to MBGNs and doped MBGNs; both ranged from 8.4 to 10.3 nm, confirming the uniformity of the mesopores. The average pore volume of doped MBGNs was similar to that of the MBGNs. The specific surface area of Sr-doped MBGNs (Sr-MBGNs) was slightly larger than that of MBGNs. However, both Zn-MBGN (210 m^2^·g^−1^) and Sr-Zn-MBGNs (180 m^2^·g^−1^) exhibited a slightly greater specific surface area compared to the control (MBGNs). These findings aligned with those of a prior investigation, in which the specific surface area increased with increasing cationic substitutions of Ca [73]. This might be because the silicate network became less compact when Ca was substituted with Sr and Zn [71]. All of these results imply that the addition of Sr and Zn led to an increase in the specific surface area of the particles while maintaining the physicochemical properties such as spherical shape, amorphous structure, and textural properties, including pore volume and pore diameter. These high specific surface areas could improve the dissolution and apatite formation rates [74]. MBGNs consist of a combination of network-forming, modifying, and intermediate oxides. The introduction of multiple cations through co-doping leads to a high degree of disorder in the structure. This disorder is primarily due to the significant flexibility in the angles between interconnected tetrahedra and their orientations. When alkali or alkaline-earth metal cations, termed “modifier” cations, are introduced into the continuous and interconnected SiO_2_ 3D network, they disrupt the silicate structure. This disruption occurs by substituting Si–BO–Si bonds with Si–NBO, leading to ionic interactions between non-bridging oxygens and modifiers. These interactions play a critical role in preserving local charge equilibrium and ensuring the overall charge neutrality [75]. Thus, multiple-cation co-doped MBGNs had significantly affected physical properties, including increased hydrodynamic diameter size (nm, DLS) and specific surface area compared to single-cation-doped MBGNs. The mesoporous structure was modified due to a reduced quantity of CTAB employed in the synthesis process compared to earlier studies [31,76]. In addition, a prior investigation suggested that an elevation in Ca content has a detrimental impact on improving the specific surface area [77]. Thus, the multiple-cation co-doped MBGNs (ternary or quaternary glass system) exhibit a greater specific surface area in comparison to the original MBGNs (binary glass system).

### 3.2. In Vitro Bioactivity

The synthesized MBGNs were immersed in the SBF for 21 days at 37 °C and 120 rpm to assess their bioactivity. After being in contact with SBF, the MBGNs formed an HCA layer on their surface through an ion exchange. That HCA layer promoted the integration of the MBGNs with living tissues. The SEM images depicted the emergence of “cauliflower-like” structures on the surfaces of the particles (Figure 5), indicating the formation of apatite deposits following 21 days of immersion in the SBF. SEM-EDS profiles illustrate the distribution of Si, Ca, Sr, Zn, and P within the particles immersed in SBF. SEM-EDS analysis confirmed the apatite formation. The Ca/P ratio was calculated using the atomic ratio [78,79]. The Ca/P atomic ratio was reported to be around 1.56–1.75, deviating slightly from the stoichiometric hydroxyapatite value of 1.67 [80]. The potential of these MBGNs to produce the apatite-like layer on their surface and undergo resorption may augment their bioactivity, facilitating the integration of the MBNGs with the host bone. The replacement of Sr^2+^ for Ca^2+^ ions within the particles accelerated the rates of dissolution, thus promoting the formation of apatite [81].

The FTIR spectra of the particles following immersion in the SBF solution for 21 days confirmed the distinctive emergence of peaks corresponding to the formation of an HCA layer on the particle surfaces (Figure 6a). Peaks between 3200 and 3600 cm^−1^, attributed to the hydroxyl groups (-OH), and between 1400 and 1600 cm^−1^, attributed to carbonate groups, indicated the substitution of carbonate ions within the hydroxyapatite layer [82]. The XRD spectra of the immersed particles further validated the presence of apatite formation (Figure 6b). The key peaks of hydroxyapatite were observed at 2θ = 22.9°, 25.9°, 31.7°, 32.2°, 34.0°, 43.8°, and 45.3° (JCPDS card No. 09–0432) [83].

### 3.3. In Vitro Cytotoxicity

The cytotoxic impact of MBGNs on hMSCs was examined in vitro. The metabolic cell viability of hMSCs treated with MBGNs, Sr-MBGNs, Zn-MBGNs, and Sr-Zn-MBGNs at 0 to 1000 µg/mL is shown in Figure 7. The MBGNs and Sr-MBGNs did not elicit toxic effects on the cellular environment up to 1000 µg/mL, as shown in Figure 7. There was a notable statistically significant reduction in the metabolic cell activity of hMSCs upon treated with Zn-MBGNs and Sr-Zn-MBGNs at ≥250 µg/mL (*p* < 0.05). The cell viability decreased progressively with higher concentrations of Zn-MBGNs and Sr-Zn-MBGNs, indicating that Zn negatively impacts the metabolic cell viability of hMSCs. Thus, 200 µg/mL is the cut-off-level particle concentration that can be used without altering cell viability. A prior investigation indicated that the accelerated release of Zn^2+^ ions from BGs led to cellular harm and mortality by triggering oxidative stress [66,84]. High Zn concentrations (20 wt%) in BGs have been shown to impede the growth of cells and HA formation [85].

### 3.4. In Vitro Mineralization

Calcium mineralization often serves as an indicator of osteoblast cell differentiation, a crucial step in the beginning of mineralization formation during bone regeneration [58]. In this study, calcified nodule formation in vitro was detected after 1, 2, and 3 weeks in culture without osteogenic supplements. After treating hMSCs with MBGNs, Sr-MBGNs, Zn-MBGNs, and Sr-Zn-MBGNs ([NPs] 200 µg/mL), clear evidence of calcification was observed, particularly in the Sr-MBGNs, Zn-MBGNs, and Sr-Zn-MBGNs compared to the negative control (untreated cell culture under standard conditions) after a two-week culture period (Figure 8a). The original MBGNs promoted mineral nodule formation after 3 weeks in culture. Moreover, the density of mineralization exhibited a significant increase in the Sr-MBGN, Zn-MBGN, and Sr-Zn-MBGN groups than that of the MBGN group. Sr-MBGNs, Zn-MBGNs, and Sr-Zn-MBGNs also exhibited mineral precipitation without osteogenic supplements. The capacity of particles to undergo mineralization under basal conditions was compared to traditional mineralization-promoting culture conditions (osteogenic condition, OST). Sr-MBGNs, Zn-MBGNs, and Sr-Zn-MBGNs demonstrated a limited ability to generate mineralized deposits compared to the positive control group (cells in the osteogenic environment without particles). The results of the semi-quantification of Ca deposition (OD_562_) were consistent with the mineralization staining (Figure 8b). During the initial week, there were no discernible distinctions among the various particle groups. Nevertheless, it became evident that Sr-MBGNs, Zn-MBGNs, and Sr-Zn-MBGNs prompted greater calcification when compared to MBGNs. This suggests that Sr-MBGNs, Zn-MBGNs, and Sr-Zn-MBGNs initiate the formation of calcium more rapidly than MBGNs. These findings validated the role of Sr and Zn in bone development [66,86]. This study observed an increase in calcium formation and cell density over time, suggesting that the differentiation of treated hMSCs was influenced by the duration of the study. These results also suggest that Sr-MBGNs, Zn-MBGNs, and Sr-Zn-MBGNs alone (without osteogenic supplements) have the ability to trigger the formation of mineralized nodules.

### 3.5. Osteogenic Differentiation

The osteogenic properties of hMSCs exposed to MBGNs, Sr-MBGNs, Zn-MBGNs, and Sr-Zn-MBGNs were investigated by measuring the expression levels of *ALP, SPARC*, *COL1A1*, and *RUNX2* using qPCR, as illustrated in Figure 9. Without the osteogenic inducers, the expression levels of *RUNX2*, *ALP*, and *SPARC* were notably and statistically upregulated in hMSCs treated with Sr-MBGNs, Zn-MBGNs, and Sr-Zn-MBGNs compared to traditional MBGNs (60SiO_2_-40CaO) following 1, 2, and 3 weeks in culture (* *p* < 0.05). Additionally, the expression level of *COL1A1* was statistically significantly increased following 2 and 3 weeks of culture. The impact of doping ions was assessed, revealing that the levels of *COL1A1*, *ALP,* and *RUNX2* in hMSCs exposed to Zn-MBGNs and Sr-Zn-MBGNs were significantly higher compared to Sr-MBGNs after 2 and 3 weeks of culture, respectively. Furthermore, the gene expression levels of *COL1A1*, *ALP*, and *RUNX2* in hMSCs exposed to Sr-Zn-MBGNs were significantly upregulated compared to Sr-MBGNs and Zn-MBGNs after 2 and 3 weeks of culture. Zn induces the bone formation process, including the differentiation of osteoblasts, collagen synthesis, the mineralization process, and bone fracture repair [87,88]. Sr has potential effects on bone formation by enhancing bone mineralization, stimulating osteoblast activity, and inhibiting osteoclast activity [89,90]. Notably, Zn^2+^ and Sr^2+^ ions released from the particles exhibited a greater promotion of bone formation. Due to their osteoinductive and osteoconductive properties, MSNGs have gained extensive utilization as nanocarriers for bone repair and regenerating periodontal tissues. Previous studies informed the ability of Sr- and Zn-containing BGNPs to accelerate osteogenesis [47,91,92]. Sr- and Zn-doped MBGNs demonstrated their capacity to enhance both the proliferation and differentiation of bone cells. The number of cells and the intensity of staining in the hMSCs treated with Sr-MBGNs, Zn-MBGNs, and Sr-Zn-MBGNs increased significantly over the study period compared to those treated with MBGNs. *RUNX2*, a crucial transcriptional regulator necessary for both the initial and late phases of osteoblast differentiation, serves as the primary transcription factor responsible for the activation of osteoblast-specific markers such as *ALP, SPARC*, and *COL1A1* during osteogenesis. *ALP* and *COL1A1*, recognized as initial indicators of the bone formation process, are expressed during the formation of the extracellular matrix (ECM) [93]. The high level of *ALP* and *COL1A1* expression promoted an early initiation of mineralization [94], while *SPARC* played a role in triggering the calcification process through the p38 signaling pathway [95]. The statistically significant upregulation observed in the osteogenic marker in this investigation indicates the occurrence of osteogenic differentiation. In the absence of osteogenic supplements, Sr-MBGNs, Zn-MBGNs, and Sr-Zn-MBGNs exhibited a positive impact by upregulating transcriptional regulators (*RUNX2*) and osteogenic marker genes, containing *ALP*, *SPARC*, and *COL1A1*. Notably, MBGNs co-doped with Sr and Zn (Sr-Zn-MBGNs) demonstrated a higher activation of gene expression compared to particles containing only Sr (Sr-MBGNs) or Zn (Zn-MBGNs).

### 3.6. Cellular Uptake

The effect of synthesized FITC-conjugated MBGNs was first investigated by assessing the cytotoxicity using an MTT assay. The cells were treated with particles at concentrations of 1, 10, 50, 100, 200, and 250 µg/mL for a duration of 24 h. The FITC-conjugated MBGNs were not toxic at any tested concentrations (Figure 10a). More importantly, the particles efficiently delivered their therapeutic cargo after uptake and internalization. Therefore, the cellular uptake was visualized using confocal fluorescence microscopy. FITC-Sr-MBGNs, FITC-Zn-MBGNs, and FITC-Sr-Zn-MBGNs were taken up and localized by the hMSCs. Green fluorescence was predominantly situated in the cytoplasm and located outside the nucleus (DAPI: blue regions), as illustrated in Figure 10b. Previous research findings have indicated that the cellular uptake and escape from endosomes exhibited a significant correlation with particle size, wherein the most favorable efficiencies were achieved with a particle size around 100 nm [96,97]. The nanoscale particles demonstrated the capacity to internalize and localize within cells while simultaneously releasing ions (Si^4+^, Ca^2+^, Sr^2+^, and Zn^2+^). Our previous studies reported that the sustained release of ions occurred through particle degradation in both cellular and buffer environments [21,98]. Taken together with the EDS-SEM analysis, it was verified that the creation of a hydroxyapatite (HA) layer was achieved through cation exchange on the particle surface. Additionally, therapeutic cations were found to be retained inside the particles after a 21-day immersion period. These (Si^4+^, Ca^2+^, Sr^2+^, and Zn^2+^) ions triggered biological activity, such as promoting the ability to form apatite, inducing the formation of mineralized nodules, and fostering antibacterial effects. Therefore, the development of MBGNs demonstrates their significant potential for utilization as nanocarriers for delivering therapeutic ions.

### 3.7. In Vitro Antibacterial Activity

Using the disk diffusion method, the synthesized MBGNs were evaluated for their antimicrobial efficacy against common Gram-negative (*Escherichia coli*) and Gram-positive (*Staphylococcus aureus*) bacteria. The positive control (1× Pen/Step) was used to verify the antimicrobial activities of the antibacterial drug. A larger zone of inhibition or zone of transparency surrounding the disks following incubation overnight implies a greater antibacterial effect. The clear zones are presented in Figure 11a,b. The results of the disk diffusion method showed non-antibacterial activity for MBGNs and presented antibacterial activity for Sr-MBGNs, Zn-MBGNs, and Sr-Zn-MBGNs against *E. coli* and *S. aureus.* The larger inhibition zones for Sr-MBGNs, Zn-MBGNs, and Sr-Zn-MBGNs against *E. coli* compared with *S. aureus* indicated a statistically significant increase in antibacterial effect against Gram-negative bacteria. The Sr-Zn-MBGNs showed the most potent antibacterial properties against both strains of bacteria. The release of Sr^2+^ and Zn^2+^ ions from the glass network influenced microbial metabolic processes, including alterations in osmolarity, the formation of reactive oxygen species (ROS), and changes in pH levels in the culture media, ultimately leading to an antibacterial effect [99,100,101]. A previous study reported that Zn^2+^ activated the generation of significant quantities of ROS within bacterial cells by interacting with the thiol group present in bacterial respiratory enzymes [102]. The finding In this study aligns with earlier research indicating that Zn has the ability to inhibit bacterial growth without affecting the growth of mammalian cells [103].

## 4. Conclusions

Spherical monodispersed Sr-doped, Zn-doped, and Sr-Zn-doped MBGNs with an average diameter size of 100 ± 20 nm and a hydrodynamic size of 160 ± 20 nm were synthesized using the modified microemulsion-assisted sol–gel process. The CTAB quantity was decreased in contrast to the conventional approach to finely adjust the specific surface area for the purpose of slowing down the dissolution rate. Moreover, the pivotal parameter for introducing ions into the network was the sequence of cation doping. The elemental composition (XRF), FTIR spectroscopy, and XRD analyses confirmed the successful incorporation of Sr and Zn in the MBGN structures. These analyses also verified that the amorphous nature of the synthesized MBGNs was maintained during the incorporation of Sr and Zn. The mesoporous characteristics of all synthesized MBGNs were confirmed through BET analysis. The incorporation of Sr and Zn did not impact the morphology of the MBGNs. All particles were biocompatible at a concentration of 200 µg/mL. The Sr- and Zn-doped MBGNs could promote the formation of an HCA layer on the surface of the particles, thereby enhancing their bioactivity. This HCA layer facilitates the integration of the nanoparticles with the surrounding bone, promoting bone regeneration and healing. Sr-MBGNs, Zn-MBGNs, and Sr-Zn-MBGNs have the capacity to enhance the formation of mineralization, thereby supporting osteoconduction. In addition, the expressions of *RUNX2*, *ALP, SPARC*, and *COL1A1* in hMSCs were upregulated by Sr-MBGNs, Zn-MBGNs, and Sr-Zn-MBGNs, signifying the promotion of osteogenic differentiation. These synthesized particles can deliver their therapeutic ions intracellularly. Moreover, these particles inhibited Gram-positive (*S. aureus*) and Gram-negative (*E. coli*) bacterial growth. Therefore, Sr-MBGNs, Zn-MBGNs, and Sr-Zn-MBGNs exhibited outstanding biological performance for osteogenic applications in bone tissue engineering. Future research is required to further investigate the synthesized MBGNs in vivo and to evaluate their osteogenic capacity.

## Figures and Tables

**Figure 1 nanomaterials-14-00575-f001:**
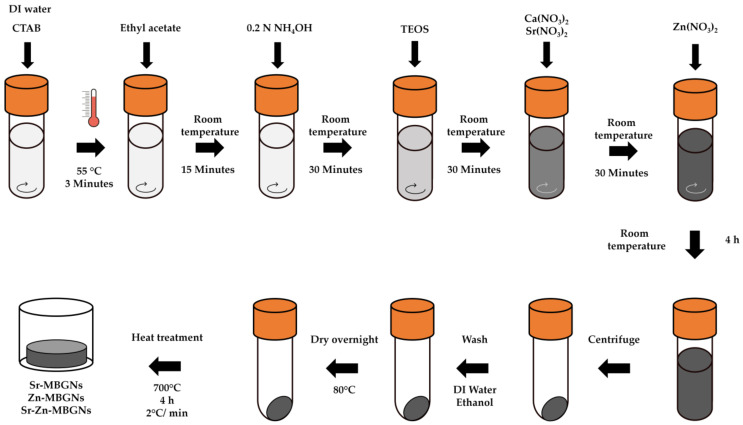
Sr− and Zn−doped MBGN synthesis by the microemulsion-assisted sol–gel process.

**Figure 2 nanomaterials-14-00575-f002:**
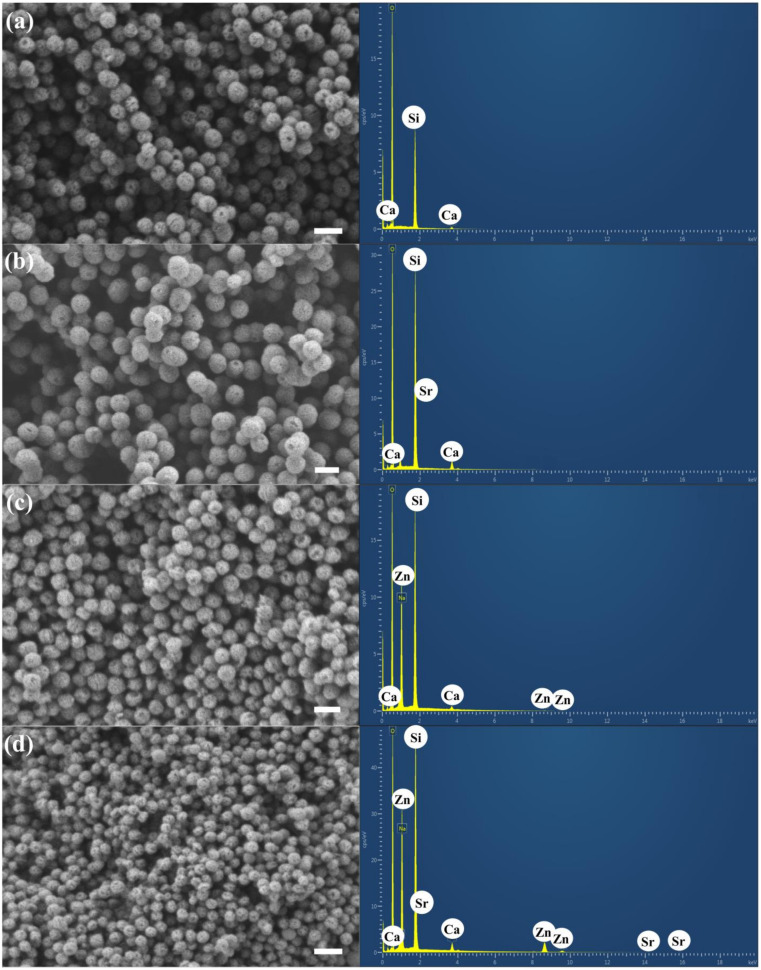
Bright-field SEM images of (**a**) MBGNs, (**b**) Sr−MBGNs, (**c**) Zn−MBGNs, and (**d**) Sr−Zn−MBGNs operating at EHT = 1 kV and 100 kX. Scale bar = 200 nm.

**Figure 3 nanomaterials-14-00575-f003:**
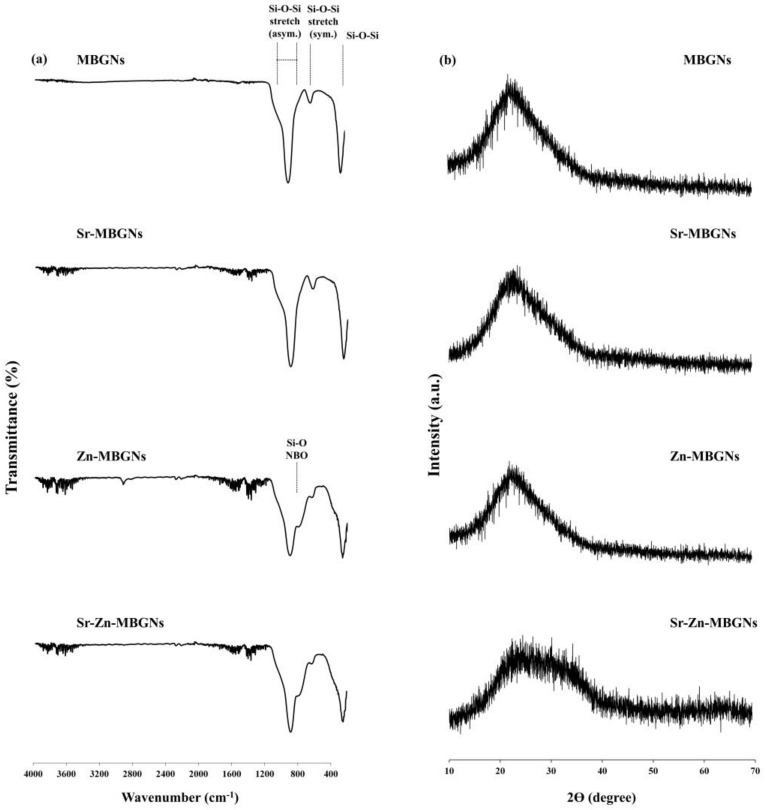
(**a**) FTIR and (**b**) XRD spectra of MBGNs, Sr−MBGNs, Zn−MBGNs, and Sr−Zn−MBGNs.

**Figure 4 nanomaterials-14-00575-f004:**
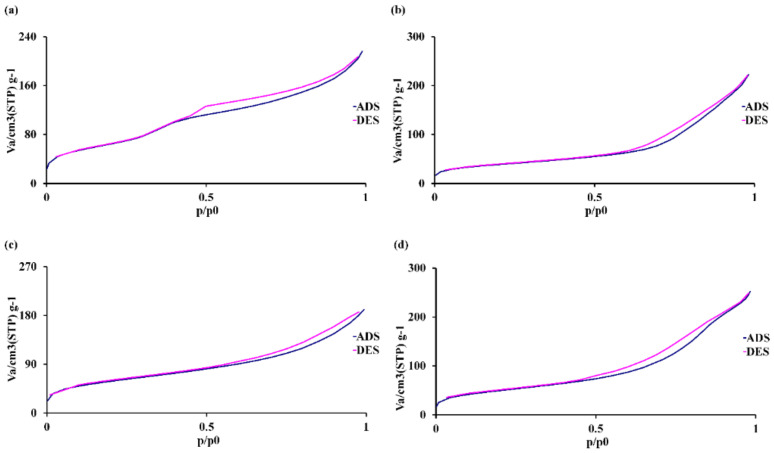
N_2_ physisorption isotherms of (**a**) MBGNs, (**b**) Sr−MBGNs, (**c**) Zn−MBGNs, and (**d**) Sr−Zn−MBGNs.

**Figure 5 nanomaterials-14-00575-f005:**
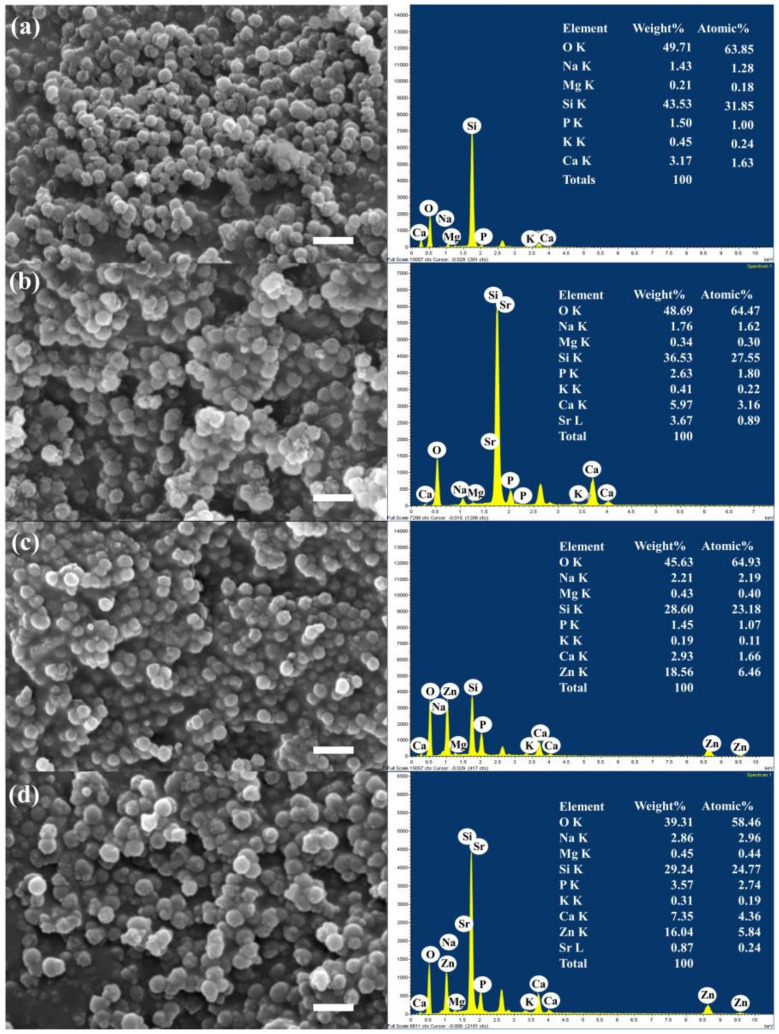
SEM images and EDS-SEM of in vitro apatite formation on (**a**) MBGNs, (**b**) Sr−MBGNs, (**c**) Zn−MBGNs, and (**d**) Sr−Zn−MBGNs after immersion in SBF for 21 days.

**Figure 6 nanomaterials-14-00575-f006:**
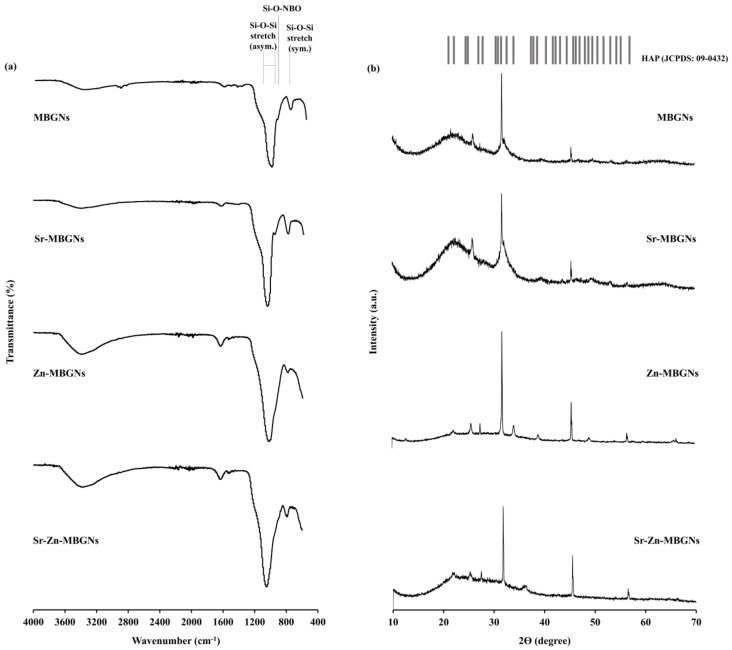
(**a**) FTIR and (**b**) XRD spectra of apatite formation on MBGNs, Sr−MBGNs, Zn−MBGNs, and Sr−Zn−MBGNs after immersion in SBF for 21 days.

**Figure 7 nanomaterials-14-00575-f007:**
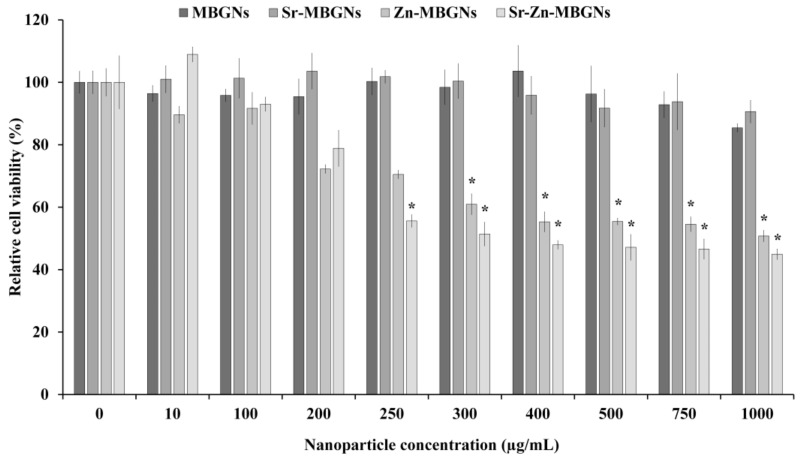
The viability of hMSCs exposed to particles was compared to the positive control (cells cultured in media without nanoparticles). Six technical replicates (*n* = 6) were conducted within each experiment, repeated in triplicate (N = 3). The data are expressed as mean ± standard deviation (SD), and statistical analysis was performed using ANOVA along with the appropriate post hoc comparison test (Tukey’s test), with significance denoted by * for *p* < 0.05.

**Figure 8 nanomaterials-14-00575-f008:**
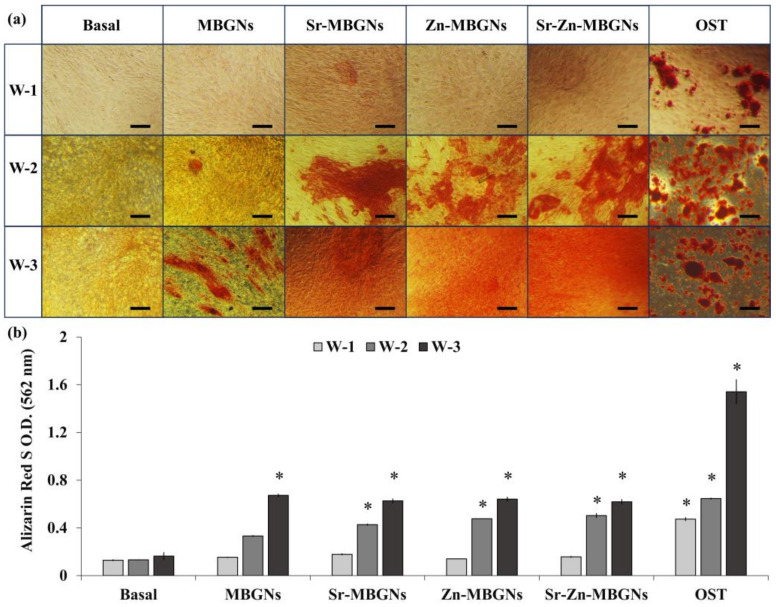
(**a**) Alizarin red S staining of hMSCs treated with MBGNs, Sr−MBGNs, Zn−MBGNs, and Sr−Zn−MBGNs at particle concentration of 200 μg/mL following 3 weeks using inverted light microscopy. Original magnification is ×10 and scale bar is 50 µm in length. (**b**) Semi-quantification of calcium deposits of hMSCs (** p* < 0.05 versus the basal control group).

**Figure 9 nanomaterials-14-00575-f009:**
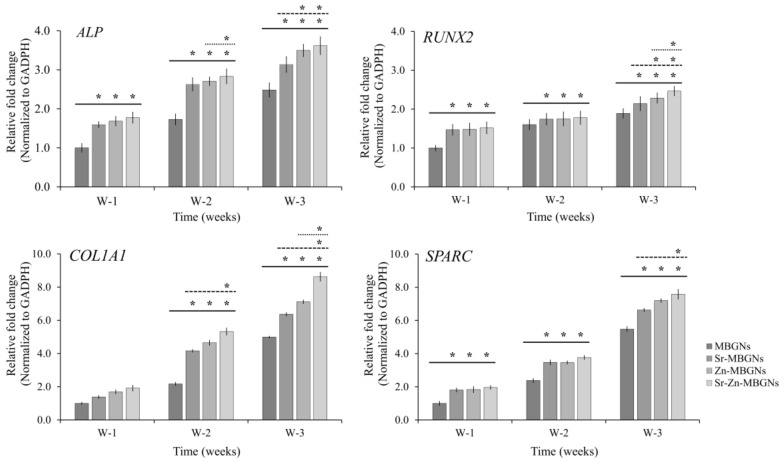
q-PCR analysis was used to assess the expression of osteogenic marker genes in hMSCs subjected to treatment with Sr−MBGNs, Zn−MBGNs, and Sr−Zn−MBGNs, all in the absence of osteogenic supplements, for 3 weeks. The quantification was performed by normalizing gene expression to GAPDH. The data presented are the results of two separate experiments, with values indicated as the mean ± SD (*n* = 3). Significant upregulation of osteogenic marker genes was observed in cells exposed to doped MBGNs compared to traditional MBGNs. (*) denotes a statistically significant difference between cells treated with doped MBGNs and traditional MBGNs under basal conditions at the corresponding time intervals (* *p* < 0.05).

**Figure 10 nanomaterials-14-00575-f010:**
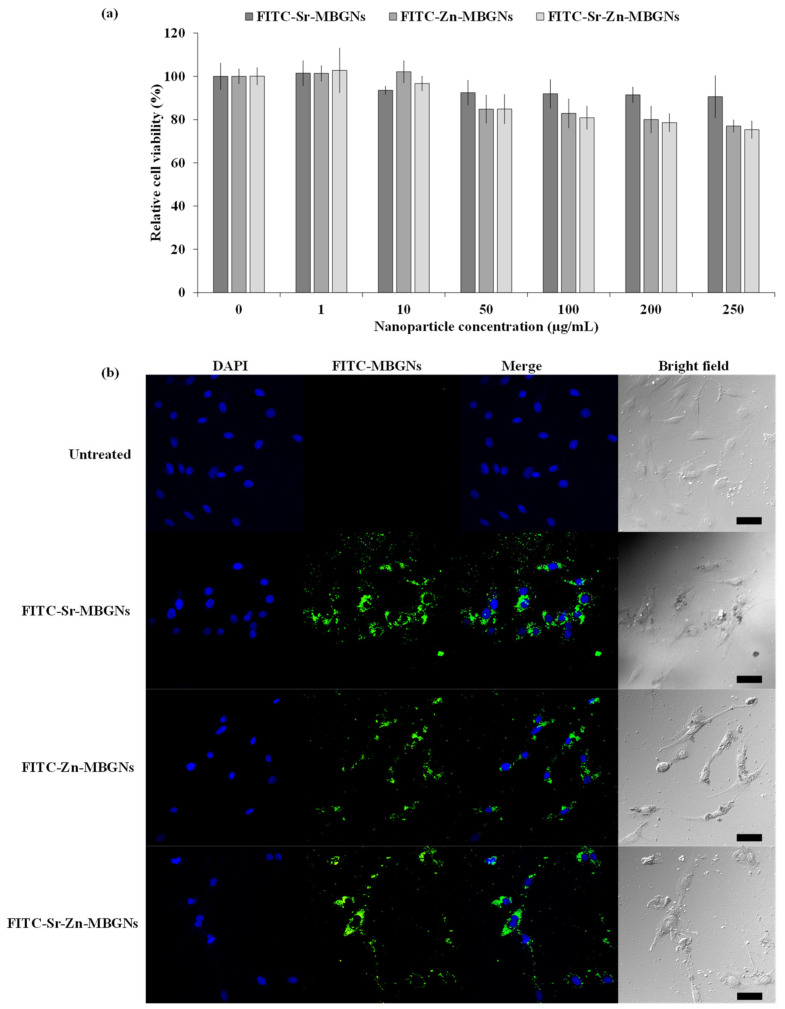
(**a**) The viability of hMSCs exposed to FITC-conjugated MBGNs was compared to the positive control (cells cultured in media without nanoparticles). Six technical replicates (*n* = 6) were conducted within each experiment, repeated in triplicate (N = 3). The data are expressed as mean ± standard deviation (SD). (**b**) Fluorescent images indicated internationalization of the FITC-conjugated MBGNs (green) at a concentration of 200 mg/mL by the hMSCs. Nuclei were counterstained with DAPI (blue). Scale bar is 50 µm in length.

**Figure 11 nanomaterials-14-00575-f011:**
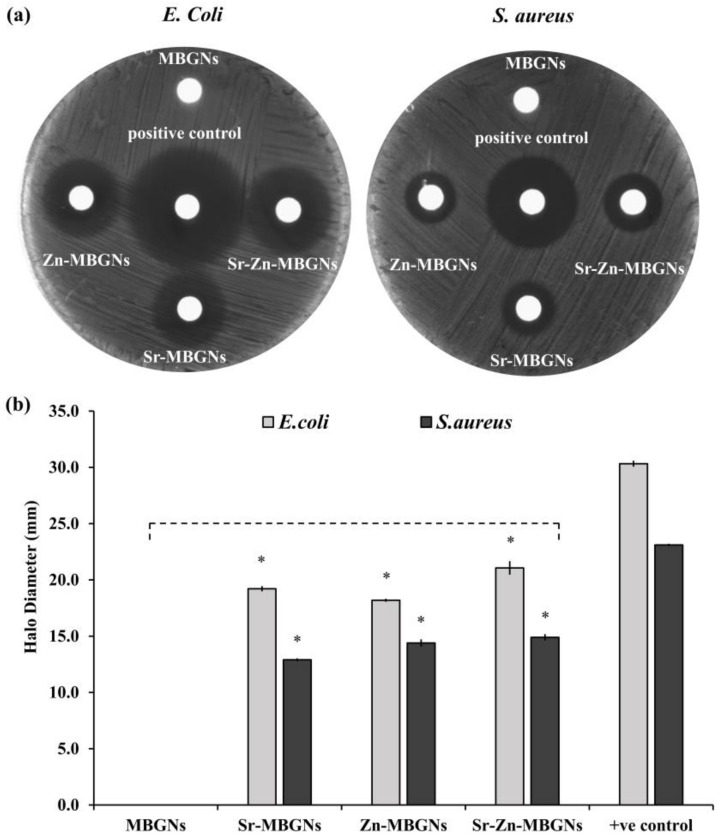
In vitro antibacterial activity (halo zone diameter) of MBGN, Sr−MBGNs, Zn−MBGNs, and Sr−Zn−MBGNs against *S. aureus* and *E. coli* after 18 h of incubation. (**a**) The different normalized widths of the antimicrobial “halo”. (**b**) Antimicrobial diffusion “halo” results: diameters of clear zones (*n* = 3). * indicating *p* < 0.05 compared to MBGNs group. Disk size was 6 mm diameter × 1 mm thickness.

**Table 1 nanomaterials-14-00575-t001:** Compositions of MBGNPs (nominal %mol).

Sample Group	%mol			
	SiO_2_	CaO	SrO	ZnO
MBGNs	60	40		
Sr-MBGNs	60	20	20	
Zn-MBGNs	60	20		20
Sr-Zn-MBGNs	60	20	10	10

**Table 2 nanomaterials-14-00575-t002:** Primers utilized for real-time q-PCR analysis.

Gene	Primer Sequences	Amplicon Size
*RUNX2*	Fw: 5′ gta gat gga cct cgg gaa cc 3′ Rw: 5′ gag gcg gtc aga gaa caa ac 3′	78 bp
*ALP*	Fw: 5′ gga act cct gac cct tga cc 3′ Rw: 5′ tcc tgt tca gct cgt act gc 3′	86 bp
*COL1A1*	Fw: 5′ gag tgc tgt ccc gtc tgc 3′ Rw: 5′ ttt ctt ggt cgg tgg gtg 3′	52 bp
*SPARC*	Fw: 5′ gag gaa acc gaa gag gag g 3′ Rw: 5′ ggg gtg ttg ttc tca tcc ag 3′	95 bp

**Table 3 nanomaterials-14-00575-t003:** Particle size and ζ potential of MBGNs.

Sample Group	Hydrodynamic Diameter Size (nm)	PI	Zeta Potential (mV)
MBGNs	142.8 ± 6.7	0.441 ± 0.095	−41.2 ± 0.6
Sr-MBGNs	161.1 ± 3.2	0.340 ± 0.037	−40.4 ± 2.1
Zn-MBGNs	188.1 ± 5.2	0.288 ± 0.007	−42.3 ± 1.9
Sr-Zn-MBGNs	181.1 ± 0.9	0.166 ± 0.025	−41.9 ± 0.8

**Table 4 nanomaterials-14-00575-t004:** Molar compositions of MBGNPs.

Sample Group	%mol			
	SiO_2_	CaO	SrO	ZnO
MBGNs	83.1 ± 0.8	16.9 ± 0.7		
Sr-MBGNs	78.3 ± 0.6	10.0 ± 0.4	11.7 ± 0.7	
Zn-MBGNs	77.0 ± 0.7	5.8 ± 0.1		17.2 ± 0.7
Sr-Zn-MBGNs	78.5 ± 0.8	4.4 ± 0.2	1.8 ± 0.3	15.3 ± 0.4

**Table 5 nanomaterials-14-00575-t005:** Texture analysis of MBGNs.

Sample Group	[m^2^ g^−1^]	[cm^3^ g^−1^]	[nm]
	a_s,BET_	Pore Volume	Pore Diameter
MBGNs	120	0.28	9.4
Sr-MBGNs	137	0.34	10.0
Zn-MBGNs	210	0.29	8.6
Sr-Zn-MBGNs	180	0.39	9.7

## Data Availability

The datasets used and/or analyzed during the current study are available from the corresponding author on reasonable request.

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
