# Peer review of "Strontium and Zinc Co-Doped Mesoporous Bioactive Glass Nanoparticles for Potential Use in Bone Tissue Engineering Applications"

_nanomaterials, 2024, doi:10.3390/nano14070575_

Round 1

Reviewer 1 Report

Comments and Suggestions for Authors

The paper by Naruphontjirakul et al. provides a very complete and detailled study on Zn/Sr doped mesoporous bioactive glass nanoparticles.

Both physico chemical and biological characterization of the materials are provided.

Globally the paper is very well written and conclusions from results are sound.

As a minor comment the values of SSA and pore size should be given with less precision as the technique does not allow such exactitude.

The sigma on measurement could be given and would atteste the previous comment.

For SSA digits are not significant , for pore size the second digit (corresponding to one tenth of Angstrom) is not significant neither.

This could be easily corrected, provided this small correction the paper can be accepted for publication and could be chosen to be highligthed by the journal.

Reviewer 2 Report

Comments and Suggestions for Authors

Article with several important revisions needed

1) smoother introduction, in the paragraph where therapeutic ions are mentioned there is a lot of confusion

2) nominal composition different from the experimental, with an unclear description of the results obtained and, above all, incorrect reference to ZnO values above the nominal value

3) table 3 are all decimals correct?

4) Figure 5 does not show HCA, the Ca/P ratios of the HCA cannot be seen

5) Figure 6 IR and XRD peaks must be shown

6) Figure 11a halo not clear

Comments on the Quality of English Language

nothing to declare

Reviewer 3 Report

Comments and Suggestions for Authors

The work is devoted to the preparation of mesoporous nanocomplexes based on MBGNs functionalized with strontium and zinc and the study of their bioactive properties with a focus on osteogenic activity. The manuscript has a good structure. The data is well presented visually and has a detailed description and analysis.

However, there are a number of questions and recommendations that would help improve this manuscript.

1. Authors should carefully check all abbreviations in the manuscript text. For example, the abbreviation SBF (probably referring to Simulated Body Fluid) is indicated without decoding in lines 22, 191,192, 196 and is deciphered in the text much later (line 446).

2. The text contains unidentified characters and text (lines 148,149; line 136).

3. Authors should carefully check the manufacturer's and country indications for each of the materials used, not for selected ones (lines 127-142).

4. Table 5. “Texture analysis of MBGNs” (line 441) there is a text overlay in the first column.

5. What is the reason for using only one specific concentration of 200 μg/ml of Sr-MBGNs, Zn-MBGNs, and Zn-Sr-MBGNs used in experiments studying hMSCs cell mineralization in vitro, as well as osteogenic defferentiation?

6. The results of other similar studies should be separated from the results obtained in this work and presented in a separate paragraph, as this may confuse the reader (line 618).

7. The authors of the work repeatedly point out in the text the effects caused by the release of zinc and strontium ions from nanocomposites (lines 116,117; 362; 456; 483; 545; 619,620). In this case, the spelling of zinc and strontium in the text should be indicated as ions rather than elements. Have the authors experimentally tested the activity of releasing zinc and strontium cations from the resulting nanocomplexes?

8. Results on the antibacterial properties of nanomaterials are of significant interest, however, it would be useful to add to the discussion the mechanisms of antibacterial activity and consistency with other works, for example, 10.3390/ma14216281

9. Osteonectin is a glycoprotein, in humans it is encoded by the SPARC gene, not Osteonectin. Check it out.

10. In the conclusions there is no mention of the results obtained on the effect of MBGNs on the expression of ALP, Osteonectin, COL1A1, and RUNX2 in hMSCs. These findings may be of some interest and also need emphasis.

Round 2

Reviewer 2 Report

Comments and Suggestions for Authors

Manuscript revisions improve the quality of the manuscript, even if it remains not totally innovative. It is not correct to leave wt% in EDS, Ca/P is a ratio in moles and not by weight.To prove HA you have to evaluate the ratio in moles as the literature has taught for years.
